# Generative Local Metric Learning for Kernel Regression

**Yung-Kyun Noh**
Seoul National University, Rep. of Korea
nohyung@snu.ac.kr

**Masashi Sugiyama**
RIKEN / The University of Tokyo, Japan
sugi@k.u-tokyo.ac.jp

**Kee-Eung Kim**
KAIST, Rep. of Korea
kekim@cs.kaist.ac.kr

**Frank C. Park**
Seoul National University, Rep. of Korea
fcp@snu.ac.kr

**Daniel D. Lee**
University of Pennsylvania, USA
ddlee@seas.upenn.edu

## Abstract

This paper shows how metric learning can be used with Nadaraya-Watson (NW) kernel regression. Compared with standard approaches, such as bandwidth selection, we show how metric learning can significantly reduce the mean square error (MSE) in kernel regression, particularly for high-dimensional data. We propose a method for efficiently learning a good metric function based upon analyzing the performance of the NW estimator for Gaussian-distributed data. A key feature of our approach is that the NW estimator with a learned metric uses information from both the global and local structure of the training data. Theoretical and empirical results confirm that the learned metric can considerably reduce the bias and MSE for kernel regression even when the data are not confined to Gaussian.

## 1 Introduction

The Nadaraya-Watson (NW) estimator has long been widely used for nonparametric regression [16, 26]. The NW estimator uses paired samples to compute a locally weighted average via a kernel function, $K(\cdot, \cdot) \colon \mathbb{R}^D \times \mathbb{R}^D \to \mathbb{R}$, where $D$ is the dimensionality of data samples. The resulting estimated output for an input $\mathbf{x} \in \mathbb{R}^D$ is given by the equation:

$$\widehat{y}(\mathbf{x}) = \frac{\sum_{i=1}^{N} K(\mathbf{x}_i, \mathbf{x}) y_i}{\sum_{i=1}^{N} K(\mathbf{x}_i, \mathbf{x})} \tag{1}$$

for data $\mathcal{D} = \{\mathbf{x}_i, y_i\}_{i=1}^{N}$ with $\mathbf{x}_i \in \mathbb{R}^D$ and $y_i \in \mathbb{R}$, and a translation-invariant kernel $K(\mathbf{x}_i, \mathbf{x}) = K((\mathbf{x} - \mathbf{x}_i)^2)$. This estimator is regarded as a fundamental canonical method in supervised learning for modeling non-linear relationships using local information. It has previously been used to interpret predictions using kernel density estimation [11], memory retrieval, decision making models [19], minimum empirical mean square error (MSE) with local weights [10, 23], and sampling-based Bayesian inference [25]. All of these interpretations utilize the fact that the estimator will asymptotically converge to the optimal $\mathbb{E}_{p(y|\mathbf{x})}[y]$ with minimum MSE given an infinite number of data samples.

However, with finite samples, the NW output $\widehat{y}(\mathbf{x})$ is no longer optimal and can deviate significantly from the true conditional expectation. In particular, the weights given along the directions of large

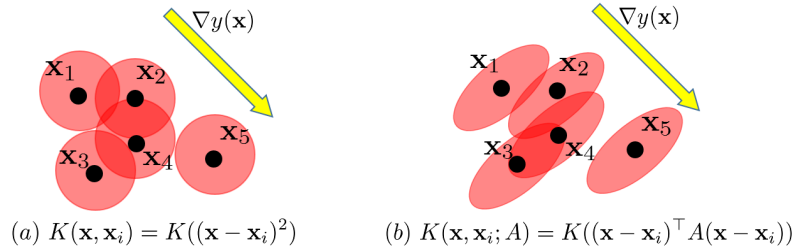

$(a)\ K(\mathbf{x}, \mathbf{x}_i) = K((\mathbf{x} - \mathbf{x}_i)^2)$      $(b)\ K(\mathbf{x}, \mathbf{x}_i; A) = K((\mathbf{x} - \mathbf{x}_i)^\top A(\mathbf{x} - \mathbf{x}_i))$

Figure 1: Metric dependency of kernels. The level curves of kernels are hyper-spheres for isotropic kernels in (a), while they are hyper-ellipsoids for kernels with the Mahalanobis metric as shown in (b). The principal directions of hyper-ellipsoids are the eigenvectors of the symmetric positive definite matrix $A$ which is used in the Mahalanobis distance. When the target variable $y$ varies along the $\nabla y$ direction in the figure, the weighted average will give less bias if the metric is extended along the orthogonal direction of $\nabla y$ as shown in (b).

variability in $y$—e.g. the direction of $\nabla y$ as in Fig. 1(a)—causes significant deviation. In this case, naively changing the kernel shape, as shown in Fig. 1(b), can alleviate the deviation. In this work, we investigate more sophisticated methods for finding appropriate kernel shapes via metric learning.

Metric learning is used to find specific directions with increased variability. Using information from the training examples, metric learning shrinks or extends distances in directions that are more or less important. A number of studies have focused on using metric learning for nearest neighbor classification [3, 6, 8, 17, 27], and many recent works have applied it to kernel methods as well [12, 13, 28]. Most of these approaches focus on modifying relative distances using triplet relationships or minimizing empirical error with some regularization.

In conventional NW regression, the deviation due to finite sampling is mitigated by controlling the bandwidth of the kernel function. The bandwidth controls the balance between the bias and the variance of the estimator, and the finite-sample deviation is reduced with appropriate selection of the bandwidth [9, 20, 21]. Other approaches include trying to explicitly subtract an estimated bias [5, 24] or using a higher-order kernel which eliminates the leading-order terms of the bias [22]. However, many of these direct approaches behave improperly in high-dimensional spaces for two reasons; distance information is dominated by noise, and by using only nearby data, local algorithms suffer due to the small number of data used effectively by the algorithms.

In this work, we apply a metric learning method for mitigating the bias. Differently from conventional metric learning methods, we analyze the metric effect on the asymptotic bias and variance of the NW estimator. Then we apply a generative model to alleviate the bias in a high-dimensional space. Our theoretical analysis shows that with a jointly Gaussian assumption on $\mathbf{x}$ and $y$, the metric learning method reduces to a simple eigenvector problem of finding a two-dimensional embedding space where the noise is effectively removed. Our approach is similar to the previous method in applying a simple generative model to mitigate the bias [18], but our analysis shows that there always exists a metric that eliminates the leading-order bias for any shape of Gaussians, and two dimensionality is enough to achieve the zero bias. The algorithm based on this analysis shows a good performance for many benchmark datasets. We interpret the result to mean that the NW estimator indirectly uses the global information through the rough generative model, and the results are improved because the information from the global covariance structure is additionally used, which would never be used in NW estimation otherwise.

One well-known extension of NW regression for reducing its bias is locally linear regression (LLR) [23]. LLR shows a zero-bias as well for data from Gaussian, but the parameter is solely estimated locally, which is prone to overfitting in high-dimensional problems. In our experiments, we compare our method with LLR and demonstrate that our method compares favorably with LLR and other competitive methods..

The rest of the paper is organized as follows. In Section 2, we explain our metric learning formulation for kernel regression. In Section 3, we derive the bias and its relationship to the metric, and our proposed algorithm is introduced in Section 4. In Section 5, we provide experiments with other standard regression methods, and conclude with a discussion in Section 6.

## 2 Metric Learning in Kernel Methods

We consider a Mahalanobis-type distance for metric learning. The Mahalanobis-type distance between two data points $\mathbf{x}_i \in \mathbb{R}^D$ and $\mathbf{x}_j \in \mathbb{R}^D$ is defined in this work as

$$||\mathbf{x}_i - \mathbf{x}_j||_A = \sqrt{(\mathbf{x}_i - \mathbf{x}_j)^\top A(\mathbf{x}_i - \mathbf{x}_j)}, \quad \left(A \succ 0, \ A^\top = A, \ |A| = 1\right) \tag{2}$$

with a symmetric positive definite matrix $A \in \mathbb{R}^{D \times D}$ and $|A|$, the determinant of A. By using this metric, we consider a metric space where the distance is extended or shrunk along the directions of eigenvectors of $A$, while the volume of the hypersphere is kept the same due to the determinant constraint. With an identity matrix $A = I$, we obtain the conventional Euclidean distance.

A kernel function capturing the local information typically decays rapidly outside a certain distance; conventionally a bandwidth parameter $h$ is used to control the effective number of data within the range of interests. If we use the Gaussian kernel as an example, with the aforementioned metric and bandwidth, the kernel function can be written as

$$K(\mathbf{x}_i, \mathbf{x}) = K\left(\frac{||\mathbf{x}_i - \mathbf{x}||_A}{h}\right) = \frac{1}{\sqrt{2\pi}^D h^D} \exp\left(-\frac{1}{2h^2}(\mathbf{x}_i - \mathbf{x})^\top A(\mathbf{x}_i - \mathbf{x})\right), \tag{3}$$

where the "relative" bandwidths along individual directions are determined by $A$, and the overall size of the kernel is determined by $h$.

## 3 Bias of Nadaraya-Watson Kernel Estimator

We first note that our target function is the conditional expectation $y(\mathbf{x}) = \mathbb{E}[y|\mathbf{x}]$, and it is invariant to metric change. When we consider a $D$-dimensional vector $\mathbf{x} \in \mathbb{R}^D$ and its stochastic prediction $y \in \mathbb{R}$, the conditional expectation $y(\mathbf{x}) = \mathbb{E}[y|\mathbf{x}]$ minimizes the MSE. If we consider two different spaces with coordinates $\mathbf{x} \in \mathbb{R}^D$ and $\mathbf{z} \in \mathbb{R}^D$ and a linear transformation between these two spaces, $\mathbf{z} = L^\top \mathbf{x}$, with a full-rank square matrix $L \in \mathbb{R}^{D \times D}$, the expectation of $y$ is invariant to the coordinate change satisfying $\mathbb{E}[y|\mathbf{x}] = \mathbb{E}[y|\mathbf{z}]$, because the conditional density is preserved by the metric change: $p(y|\mathbf{x}) = p(y|\mathbf{z})$ for all corresponding $\mathbf{x}$ and $\mathbf{z}$, and

$$\mathbb{E}[y|\mathbf{x}] = \int y \, p(y|\mathbf{x})dy = \int y \, p(y|\mathbf{z})dy = \mathbb{E}[y|\mathbf{z}]. \tag{4}$$

The equivalence means that the target function is invariant to metric change with $A = LL^\top$, and considering that the NW estimator achieves the optimal prediction $\mathbb{E}[y|\mathbf{x}]$ with infinite data, optimal prediction is achieved with infinite data regardless of the choice of metric. Thus the metric dependency is actually a finite sampling effect along with the bias and the variance.

### 3.1 Metric Effects on Bias

The bias is the expected deviation of the estimator from the true mean of the target variable $y(\mathbf{x})$:

$$\text{Bias} \quad = \quad \mathbb{E}\left[\widehat{y}(\mathbf{x}) - y(\mathbf{x})\right] = \mathbb{E}\left[\frac{\sum_{i=1}^{N} K(\mathbf{x}_i, \mathbf{x})y_i}{\sum_{i=1}^{N} K(\mathbf{x}_i, \mathbf{x})} - y(\mathbf{x})\right]. \tag{5}$$

Standard methods for calculating the bias assume asymptotic concentration around the means, both in the numerator and in the denominator of the NW estimator. Usually, the numerator and denominator of the bias are approximated separately, and the bias of the whole NW estimator is calculated using a plug-in method [15, 23]. We assume a kernel satisfying $\int K(\mathbf{z})d\mathbf{z} = 1$, $\int \mathbf{z}K(\mathbf{z})d\mathbf{z} = 0$, and $\int \mathbf{z}\mathbf{z}^\top K(\mathbf{z})d\mathbf{z} = I$. For example, the Gaussian kernel in Eq. (3) satisfies all of these conditions. Then we can first approximate the denominator as[1]

$$\mathbb{E}_{\mathbf{x}_1,\ldots,\mathbf{x}_N}\left[\frac{1}{N}\sum_{i=1}^{N} K(\mathbf{x}_i, \mathbf{x})\right] = p(\mathbf{x}) + \frac{h^2}{2}\nabla^2 p(\mathbf{x}) + \mathcal{O}(h^4), \tag{6}$$

with Laplacian $\nabla^2$, the trace of the Hessian with respect to $\mathbf{x}$. Similarly, the expectation of the numerator becomes

$$\mathbb{E}_{\substack{\mathbf{x}_1, \ldots, \mathbf{x}_N, \\ y_1, \ldots, y_N}} \left[ \frac{1}{N} \sum_{i=1}^N K(\mathbf{x}, \mathbf{x}_i) y_i \right] = p(\mathbf{x}) y(\mathbf{x}) + \frac{h^2}{2} \nabla^2 [p(\mathbf{x}) y(\mathbf{x})] + \mathcal{O}(h^4). \tag{7}$$

Using the plug-ins of Eq. (6) and Eq. (7), we can find the leading-order terms of the NW estimation, and the bias of the NW estimator can be obtained as follows:

$$\mathbb{E}\left[ \frac{\sum_{i=1}^N K(\mathbf{x}, \mathbf{x}_i) y_i}{\sum_{i=1}^N K(\mathbf{x}, \mathbf{x}_i)} - y(\mathbf{x}) \right] = h^2 \left( \frac{\nabla^\top p(\mathbf{x}) \nabla y(\mathbf{x})}{p(\mathbf{x})} + \frac{\nabla^2 y(\mathbf{x})}{2} \right) + \mathcal{O}(h^4). \tag{8}$$

Here, all gradients $\nabla$ and Laplacians $\nabla^2$ are with respect to $\mathbf{x}$. We have noted that the target $y(\mathbf{x}) = \mathbb{E}[y|\mathbf{x}]$ is invariant to the metric change, and the metric dependency comes from the finite sample deviation terms. Here, both the gradient and the Laplacian in the deviation are dependent on the change of metric $A$.

## 3.2 Conventional Methods of Reducing Bias

Previously, there have been works intended to reduce the deviation [9, 20, 21]. A standard approach is to adapt the size of bandwidth parameter $h$ under the minimum MSE criterion. Bandwidth selection has an intuitive motivation of balancing the tradeoff between the bias and the variance; the bias can be reduced by using a small bandwidth but at the cost of increasing the variance. Therefore, for bandwidth selection, the bias and variance criteria have to be used at the same time.

Another straightforward and well-known extension of the NW estimator is the locally linear regression (LLR) [2, 23]. Considering that Eq. (1) is the solution minimizing the *local* empirical MSE:

$$y(\mathbf{x}) = \arg \min_{\alpha \in \mathbb{R}} \sum_{i=1}^N (y_i - \alpha)^2 K(\mathbf{x}_i, \mathbf{x}), \tag{9}$$

the LLR extends this objective function to

$$[y(\mathbf{x}), \beta^*(\mathbf{x})] = \arg \min_{\alpha \in \mathbb{R}, \beta \in \mathbb{R}^D} \sum_{i=1}^N \left( y_i - \alpha - \beta^\top (\mathbf{x}_i - \mathbf{x}) \right)^2 K(\mathbf{x}_i, \mathbf{x}), \tag{10}$$

to eliminate the noise produced by the linear component of the target function. The vector parameter $\beta^*(\mathbf{x}) \in \mathbb{R}^D$ is the estimated local gradient using local data, and this vector often overfits in a high-dimensional space resulting in a poor solution of $\alpha$.

However, LLR asymptotically produces the bias of

$$\text{Bias}_{\text{LLR}} = \frac{h^2}{2} \nabla^2 y(\mathbf{x}) + \mathcal{O}(h^4). \tag{11}$$

Eq. (11) can be compared with the NW bias in Eq. (8), where the bias term from the linear variation of $y$ with respect to $\mathbf{x}$, $h^2 \frac{\nabla^\top p \nabla y}{p}$, is eliminated.

# 4 Metric for Nadaraya-Watson Regression

In this section, we propose a metric that appropriately reduces the metric-dependent bias of the NW estimator.

## 4.1 Nadaraya-Watson Regression for Gaussian

In order to obtain a metric, we first provide the following theorem which guarantees the existence of a good metric that eliminates the leading order bias at any point regardless of the configuration of Gaussian.

**Theorem** 1: *At any point $\mathbf{x}$, there exists a metric matrix $A$, such that for data $\mathbf{x} \in \mathbb{R}^D$ and the output $y \in \mathbb{R}$ jointly generated from any $(D+1)$-dimensional Gaussian, the NW regression with distance $d(\mathbf{x}, \mathbf{x}') = ||\mathbf{x} - \mathbf{x}'||_A$, for $\mathbf{x}, \mathbf{x}' \in \mathbb{R}^D$, has a zero leading-order bias.*

Based on the theorem, we will consider using the corresponding metric space for NW regression at each point. The theorem is proven using the following Proposition 2 and Lemma 3, which are general claims without the Gaussian assumptions.

**Proposition** 2: *There exists a symmetric positive definite matrix $A$ that eliminates the first term $\frac{\nabla^\top p(\mathbf{x}) \nabla y(\mathbf{x})}{p(\mathbf{x})}$ inside the bias in Eq. (8), when used with the metric in Eq. (2), and when there exist two linearly independent gradients of $p(\mathbf{x})$ and $y(\mathbf{x})$, and $p(\mathbf{x})$ is away from zero.*

**Proof**: We consider a coordinate transformation $\mathbf{z} = L^\top \mathbf{x}$ with $L$ satisfying $A = LL^\top$. The gradient of a differentiable function $y(.)$ and a density function $p(.)$ with respect to $\mathbf{z}$ is

$$\nabla_{\mathbf{z}} y(\mathbf{z}) \Big|_{\mathbf{z} = L^\top \mathbf{x}} = L^{-1} \nabla_{\mathbf{x}} y(\mathbf{x}) , \quad \nabla_{\mathbf{z}} p(\mathbf{z}) \Big|_{\mathbf{z} = L^\top \mathbf{x}} = \frac{1}{|L|} L^{-1} \nabla_{\mathbf{x}} p(\mathbf{x}), \tag{12}$$

and the scalar $\nabla^\top p(\mathbf{x}) \nabla y(\mathbf{x})$ in the Euclidean space can be rewritten in the transformed space as

$$
\begin{aligned}
\nabla_{\mathbf{z}}^\top p(\mathbf{z}) \nabla_{\mathbf{z}} y(\mathbf{z}) &= \frac{1}{2} \left( \nabla_{\mathbf{z}}^\top p(\mathbf{z}) \nabla_{\mathbf{z}} y(\mathbf{z}) + \nabla_{\mathbf{z}}^\top y(\mathbf{z}) \nabla_{\mathbf{z}} p(\mathbf{z}) \right) \tag{13} \\
&= \frac{1}{2|L|} \left( \nabla_{\mathbf{x}}^\top p(\mathbf{x}) L^{-\top} L^{-1} \nabla_{\mathbf{x}} y(\mathbf{x}) + \nabla_{\mathbf{x}} y(\mathbf{x}) L^{-\top} L^{-1} \nabla_{\mathbf{x}}^\top p(\mathbf{x}) \right) \tag{14} \\
&= \frac{1}{2|A|^{\frac{1}{2}}} tr \left[ A^{-1} \left( \nabla_{\mathbf{x}} y(\mathbf{x}) \nabla_{\mathbf{x}}^\top p(\mathbf{x}) + \nabla_{\mathbf{x}} p(\mathbf{x}) \nabla_{\mathbf{x}}^\top y(\mathbf{x}) \right) \right] . \tag{15}
\end{aligned}
$$

The symmetric matrix $B = \nabla y(\mathbf{x}) \nabla^\top p(\mathbf{x}) + \nabla p(\mathbf{x}) \nabla^\top y(\mathbf{x})$ has rank two with independent $\nabla y(\mathbf{x})$ and $\nabla p(\mathbf{x})$ and can be eigen-decomposed as

$$B = \begin{bmatrix} \mathbf{u}_1 & \mathbf{u}_2 \end{bmatrix} \begin{pmatrix} \lambda_1 & 0 \\ 0 & \lambda_2 \end{pmatrix} \begin{bmatrix} \mathbf{u}_1 & \mathbf{u}_2 \end{bmatrix}^\top \tag{16}$$

with eigenvectors $\mathbf{u}_1$ and $\mathbf{u}_2$ and nonzero eigenvalues $\lambda_1$ and $\lambda_2$. A sufficient condition for the existence of $A$ is that the two eigenvalues have different signs, in other words, $\lambda_1 \lambda_2 < 0$.

Let $\lambda_1 > 0$ and $\lambda_2 < 0$ without loss of generality, and we choose a positive definite matrix having the following eigenvector decomposition:

$$A = \begin{bmatrix} \mathbf{u}_1 & \mathbf{u}_2 \cdots \end{bmatrix} \begin{pmatrix} \lambda_1 & 0 & \cdots \\ 0 & -\lambda_2 & \\ \vdots & & \ddots \end{pmatrix} \begin{bmatrix} \mathbf{u}_1 & \mathbf{u}_2 \cdots \end{bmatrix}^\top . \tag{17}$$

Then Eq. (15) becomes zero, yielding a zero value for the first term of the bias with nonzero $p(\mathbf{x})$. Therefore, we can always find $A$ that eliminates the first term of the bias once $B$ has one positive and one negative eigenvalue, and the following Lemma 3 proves that $B$ always has one positive and one negative eigenvalue. ∎

**Lemma** 3: *A symmetric matrix $B = (B' + B'^\top)/2$ has two nonzero eigenvalues for a rank one matrix $B' = \mathbf{v}_1 \mathbf{v}_2^\top$ with two linearly independent vectors, $\mathbf{v}_1$ and $\mathbf{v}_2$. Here, one of the two eigenvalues is positive, and the other is negative.*

**Proof**: We can reformulate $B$ as

$$B = \frac{1}{2} (\mathbf{v}_1 \mathbf{v}_2^\top + \mathbf{v}_2 \mathbf{v}_1^\top) = \frac{1}{2} \begin{bmatrix} \mathbf{v}_1 & \mathbf{v}_2 \end{bmatrix} \begin{pmatrix} 0 & 1 \\ 1 & 0 \end{pmatrix} \begin{bmatrix} \mathbf{v}_1 & \mathbf{v}_2 \end{bmatrix}^\top . \tag{18}$$

If we make a new square matrix of size two, $M = \begin{bmatrix} \mathbf{v}_1 & \mathbf{v}_2 \end{bmatrix}^\top B \begin{bmatrix} \mathbf{v}_1 & \mathbf{v}_2 \end{bmatrix}$, the determinant of the matrix is as follows using the eigen-decomposition of $B$ with eigenvectors $\mathbf{u}_1$ and $\mathbf{u}_2$ and eigenvalues $\lambda_1$ and $\lambda_2$:

$$
\begin{aligned}
|M| &= \left| \begin{bmatrix} \mathbf{v}_1 & \mathbf{v}_2 \end{bmatrix}^\top B \begin{bmatrix} \mathbf{v}_1 & \mathbf{v}_2 \end{bmatrix} \right| \tag{19} \\
&= \left| \begin{bmatrix} \mathbf{v}_1 & \mathbf{v}_2 \end{bmatrix}^\top \begin{bmatrix} \mathbf{u}_1 & \mathbf{u}_2 \end{bmatrix} \begin{bmatrix} \lambda_1 & 0 \\ 0 & \lambda_2 \end{bmatrix} \begin{bmatrix} \mathbf{u}_1 & \mathbf{u}_2 \end{bmatrix}^\top \begin{bmatrix} \mathbf{v}_1 & \mathbf{v}_2 \end{bmatrix} \right| \tag{20} \\
&= \lambda_1 \lambda_2 \left( \mathbf{v}_1^\top \mathbf{u}_1 \mathbf{v}_2^\top \mathbf{u}_2 - \mathbf{v}_1^\top \mathbf{u}_2 \mathbf{v}_2^\top \mathbf{u}_1 \right)^2 , \tag{21}
\end{aligned}
$$

and at the same time, $|M|$ is always negative by the following derivation:

$$|M| = \left| \begin{bmatrix} \mathbf{v}_1 & \mathbf{v}_2 \end{bmatrix}^\top B \begin{bmatrix} \mathbf{v}_1 & \mathbf{v}_2 \end{bmatrix} \right| = \frac{1}{2} \left| \begin{bmatrix} \mathbf{v}_1 & \mathbf{v}_2 \end{bmatrix}^\top \begin{bmatrix} \mathbf{v}_1 & \mathbf{v}_2 \end{bmatrix} \right|^2 \left| \begin{pmatrix} 0 & 1 \\ 1 & 0 \end{pmatrix} \right| < 0. \tag{22}$$

From these calculations, $\lambda_1 \lambda_2 < 0$, and $\lambda_1$ and $\lambda_2$ always have different signs. ∎

With Proposition 2 and Lemma 3, we always have a metric space associated with $A$ in Eq. (17) that eliminates the leading order bias of a Gaussian, because $\nabla^2 y(\mathbf{x}) = 0$ is always satisfied for $\mathbf{x}$ and $y$ which are jointly Gaussian, eliminating the second term of Eq. (8) as well.

## 4.2 Gaussian Model for Metric Learning

We now know there exists an interesting scaling by a metric change where the NW regression achieves the bias $\mathcal{O}(h^4)$. The metric we use is as follows:

$$A_{\mathrm{NW}} = \beta [\mathbf{u}_+ \mathbf{u}_-] \begin{pmatrix} \lambda_+ & 0 \\ 0 & -\lambda_- \end{pmatrix} [\mathbf{u}_+ \mathbf{u}_-]^\top + \gamma I, \qquad \text{for} \quad |A_{\mathrm{NW}}| = 1. \tag{23}$$

Here, $\beta$ is the constant determined from the constraint $|A_{\mathrm{NW}}| = 1$. We use one positive and one negative eigenvalue, $\lambda_+ > 0$ and $\lambda_- < 0$, from matrix $B$:

$$B = \nabla y(\mathbf{x}) \nabla^\top p(\mathbf{x}) + \nabla p(\mathbf{x}) \nabla^\top y(\mathbf{x}), \tag{24}$$

and their corresponding eigenvectors $\mathbf{u}_+$ and $\mathbf{u}_-$. A small positive regularization constant $\gamma$ is added after being multiplied by the identity matrix.

By adding a regularization term to the metric, the deviation with exact $\nabla p(\mathbf{x})$ and $\nabla y(\mathbf{x})$ becomes nonzero, but a small value, $\frac{h^2}{2p(\mathbf{x})} \mathrm{tr}[A_{\mathrm{NW}}^{-1} B] = \frac{h^2}{2p(\mathbf{x})\beta} \left( \frac{\lambda_+}{\lambda_+ + \gamma} - \frac{\lambda_-}{\lambda_- + \gamma} \right) = \frac{\gamma h^2}{2p(\mathbf{x})\beta} \left( \frac{\lambda_+ - \lambda_-}{\lambda_+ \lambda_-} \right) + \mathcal{O}(\gamma^2)$. However, with small $\gamma$, the deviation is still low unless $p(\mathbf{x})$ is close to zero, or $\nabla p(\mathbf{x})$ and $\nabla y(\mathbf{x})$ are parallel.

The matrix $A_{\mathrm{NW}}$ is obtained for every point of interest, and the NW regression of each point is performed with a different $A_{\mathrm{NW}}$ calculated at each point. $A_{\mathrm{NW}}$ is a function of $\mathbf{x}$, but the changing part is only the rank two matrix, and the calculation is simple, since we only have to solve the eigenvector problem of a $2 \times 2$ matrix for each query point regardless of the original dimensionality. Note that the bandwidth $h$ is not yet included for the optimization when we obtain the metric. After we obtain the metric, we can still use bandwidth selection for even better MSE.

In order to obtain the metric $A_{\mathrm{NW}}$, at every query, we need the information of $\nabla p(\mathbf{x})$ and $\nabla y(\mathbf{x})$. The knowledge of true $y(\mathbf{x})$ and $p(\mathbf{x})$ is unknown, and we need to obtain the gradient information from data again. Previously, the gradient information was obtained locally with a small number of samples [4, 7], but such methods are not preferred here because we need to overcome the corruption of the local information in high-dimensional cases. Instead, we use a global parametric model: Using a single Gaussian model for all data, we estimate the gradient of true $y(\mathbf{x})$ and $p(\mathbf{x})$ at each point from the global configuration of data fitted by a single Gaussian:

$$p\left( \begin{pmatrix} y \\ \mathbf{x} \end{pmatrix} \right) = \mathcal{N}\left( \begin{pmatrix} \mu_y \\ \mu_{\mathbf{x}} \end{pmatrix}, \begin{pmatrix} \Sigma_y & \Sigma_{y\mathbf{x}} \\ \Sigma_{\mathbf{x}y} & \Sigma_{\mathbf{x}} \end{pmatrix} \right). \tag{25}$$

In fact, the target function $y(\mathbf{x}) = \Sigma_{y\mathbf{x}} \Sigma_{\mathbf{x}}^{-1} (\mathbf{x} - \mu_{\mathbf{x}}) + \mu_y$ (See Appendix) can be analytically obtained in a closed form when we estimate the parameters of the Gaussian, but we reuse $y(\mathbf{x})$ for enhancement of the NW regression, and the NW regression updates $y(\mathbf{x})$ using local information. The gradients for metric learning can be obtained using $\nabla y(\mathbf{x}) = \widehat{\Sigma}_{\mathbf{x}}^{-1} \widehat{\Sigma}_{\mathbf{x}y}$ and $\frac{\nabla p(\mathbf{x})}{p(\mathbf{x})} = -\widehat{\Sigma}_{\mathbf{x}}^{-1} (\mathbf{x} - \widehat{\mu}_{\mathbf{x}})$ from the estimated parameters $\widehat{\Sigma}_{\mathbf{x}}$, $\widehat{\Sigma}_{\mathbf{x}y}$, and $\widehat{\mu}_{\mathbf{x}}$ if the global model is Gaussian. A pseudo-code of the proposed method is presented in Algorithm 1.

## 4.3 Interpretation of the Metric

The learned metric $A_{\mathrm{NW}}$ considers the two-dimensional subspace spanned by $\nabla p(\mathbf{x}) = -p(\mathbf{x}) \Sigma_{\mathbf{x}}^{-1} (\mathbf{x} - \mu_{\mathbf{x}})$ and $\nabla y(\mathbf{x}) = \Sigma_{\mathbf{x}}^{-1} \Sigma_{\mathbf{x}y}$. The two-dimensionality analysis of the metric shows that the distant points are used for those in the space orthogonal to this two-dimensional subspace.

**Algorithm 1** Generative Local Metric Learning for NW Regression

---

**Input:** data $\mathcal{D} = \{\mathbf{x}_i, y_i\}_{i=1}^{N}$ and point for regression $\mathbf{x}$
**Output:** regression output $\widehat{y}(\mathbf{x})$
**Procedure:**

1: Find joint covariance matrix $\Sigma = \begin{pmatrix} \Sigma_y & \Sigma_{y\mathbf{x}} \\ \Sigma_{\mathbf{x}y} & \Sigma_{\mathbf{x}} \end{pmatrix}$ and mean vector $\mu = \begin{pmatrix} \mu_y \\ \mu_{\mathbf{x}} \end{pmatrix}$ from data $\mathcal{D}$.

2: Obtain two eigenvectors

$$\mathbf{u}_1 = \frac{\nabla p(\mathbf{x})}{||\nabla p(\mathbf{x})||} + \frac{\nabla y}{||\nabla y||} \quad \text{and} \quad \mathbf{u}_2 = \frac{\nabla p(\mathbf{x})}{||\nabla p(\mathbf{x})||} - \frac{\nabla y}{||\nabla y||}, \tag{26}$$

and their corresponding eigenvalues

$$\lambda_1 = \frac{1}{2p(\mathbf{x})}(\nabla y^\top \nabla p + ||\nabla y|| ||\nabla p||) \quad \text{and} \quad \lambda_2 = \frac{1}{2p(\mathbf{x})}(\nabla y^\top \nabla p - ||\nabla y|| ||\nabla p||), \tag{27}$$

using

$$\nabla p(\mathbf{x}) = -p(\mathbf{x})\Sigma_{\mathbf{x}}^{-1}(\mathbf{x} - \mu_{\mathbf{x}}) \quad \text{and} \quad \nabla y = \Sigma_{\mathbf{x}}^{-1}\Sigma_{\mathbf{x}y}. \tag{28}$$

3: Obtain the transform matrix $L$ using $\mathbf{u}_1$, $\mathbf{u}_2$, $\lambda_1$, and $\lambda_2$:

$$L = \begin{pmatrix} | & | & \\ \frac{\mathbf{u}_1}{||\mathbf{u}_1||} & \frac{\mathbf{u}_2}{||\mathbf{u}_2||} & U_o \\ | & | & \end{pmatrix} \begin{pmatrix} \sqrt{\lambda_1 + \gamma}/T & & & & \\ & \sqrt{-\lambda_2 + \gamma}/T & & & \\ & & \sqrt{\gamma}/T & & \\ & & & \sqrt{\gamma}/T & \\ & & & & \ddots & \\ & & & & & \sqrt{\gamma}/T \end{pmatrix} \tag{29}$$

with $T = \left((\lambda_1 + 1)(-\lambda_2 + \gamma)\gamma^{D-2}\right)^{\frac{1}{2D}}$, a small constant $\gamma$, and an orthonomal matrix $U_o \in \mathbb{R}^{D \times (D-2)}$ spanning the normal space of $\mathbf{u}_1$ and $\mathbf{u}_2$.

4: Perform NW regression at $\mathbf{z} = L^\top \mathbf{x}$ using transformed data $\mathbf{z}_i = L^\top \mathbf{x}_i, i = 1, \ldots, N$.

---

This fact has the effect of virtually increasing the amount of data compared with algorithms with isotropic kernels, particularly in high-dimensional space.

The following proposition gives an intuitive explanation that the bias reduction is more important in high-dimensional space than the reduction of the variance once the optimal bandwidth has been selected balancing the leading terms of the bias and variance after the change of metric. Proposition 2, Lemma 3, and the following Proposition 4 are obtained without any Gaussian assumption.

**Proposition** 4: *Let us simplify the MSE as the squared bias obtained from the leading terms in Eq. (8) and the variance[2], i.e.,*

$$f(h) = h^4 C_1 + \frac{1}{Nh^D}C_2. \tag{31}$$

*Then, at some $h^*$, it has the the minimum $f(h_*) = C_1$ in the limit with infinite $D$, where $D$ is the dimensionality of data.*

**Proof**: The optimal $h$ can be obtained using $\left.\frac{\partial f(h)}{\partial h}\right|_{h=h_*} = 0$, and the optimal $h$ is

$$h_* = N^{-\frac{1}{D+4}}\left(\frac{D \cdot C_2}{4 \cdot C_1}\right)^{\frac{1}{D+4}}. \tag{32}$$

$$C_1 = \left(\frac{\nabla^\top p(\mathbf{x})\nabla y(\mathbf{x})}{p(\mathbf{x})} + \frac{\nabla^2 y(\mathbf{x})}{2}\right)^2 \quad and \quad C_2 = \frac{1}{(2\sqrt{\pi})^D}\frac{\sigma_y^2(\mathbf{x})}{p(\mathbf{x})} \tag{30}$$

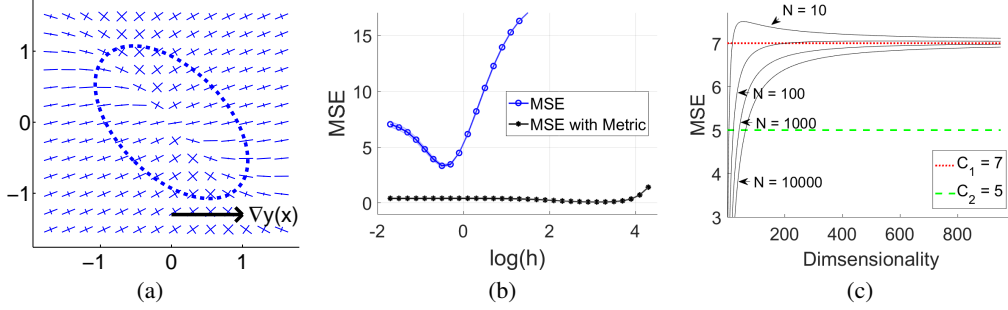

Figure 2: (a) Metric calculation for a Gaussian and gradient $\nabla y$. (b) Empirical MSEs with and without the metric. (c) Leading order terms in MSE with optimal bandwidth for various numbers of data.

By plugging $h_*$ into $f(h)$ in Eq. (31), we obtain

$$f(h_*) = N^{-\frac{4}{D+4}} \left( \left( \frac{D}{4} \right)^{\frac{4}{D+4}} + \left( \frac{4}{D} \right)^{\frac{D}{D+4}} \right) C_1^{\frac{D}{D+4}} C_2^{\frac{4}{D+4}} \simeq C_1. \quad \text{(for } D \gg 4). \ \blacksquare \quad (33)$$

In Proposition 4, the first term $h^4 C_1$ is the square of the bias, and the second term $\frac{1}{Nh^D} C_2$ is the derived variance. The MSE is minimized in a high-dimensional space only through the minimization of the bias when it is accompanied by the optimization with respect to the bandwidth $h$. The plot of MSE in Fig. 2(c) shows that the MSE with bandwidth selection quickly approaches $C_1$ in particular with a small number of data. The derivation shows that we can ignore the variance optimization with respect to the metric change. We only focus on achieving a small bias and rather than minimizing the variance, the bandwidth selection follows later.

## 5 Experiments

The proposed algorithm is evaluated using both synthetic and real datasets. For a Gaussian, Fig. 2(a) depicts the eigenvectors along with the eigenvalues of the matrix $B = \nabla y \nabla^\top p + \nabla p \nabla^\top y$ at different points in the two-dimensional subspace spanned by $\nabla y$ and $\nabla p$. The metric can be compared with the adaptive scaling proposed in [14], which determines the metric according to the average amount of $\nabla y$. Our metric also uses $\nabla y$, but the metric is determined using the relationship with $\nabla p$.

Fig. 2(a) shows the metric eigenvalues and eigenvectors at each point for a two-dimensional Gaussian with a covariance contour in the figure. With Gaussian data, the MSE with the proposed metric is shown along with MSE with the Euclidean metric in Fig. 2(b). The metric is obtained from the estimated parameter of a jointly Gaussian model, where the result with a learned metric shows a huge difference in the MSE.

For real-data experiments, we used the Delve datasets (Abalone, Bank-8fm, Bank-32fh, CPU), UCI datasets (Community, NavalC, NavalT, Protein, Slice), KEEL datasets (Ailerons, Elevators, Puma32h) [1], and datasets from a previous paper (Pendulum, Pol) [15]. The datasets include dozens of features and several thousands to tens of thousands of data. Using a Gaussian model with regularized maximum likelihood estimated parameters, we apply a metric which minimizes the bias with a fixed $\gamma = \max(|\lambda_1|, |\lambda_2|) \times 10^{-2}$, and we choose $h$ from a pre-chosen validation set. NW estimation with the proposed metric (NW+GMetric) is compared with the conventional NW estimation (NW), LLR (LLR), the previous metric learning method for NW regression (NW+WMetric [28], NW+KMetric [14]), a more flexible Gaussian process regression (GPR) with the Gaussian kernel, and the Gaussian globally linear model (GGL) using $y(\mathbf{x}) = \widehat{\Sigma}_{y\mathbf{x}} \widehat{\Sigma}_{\mathbf{x}}^{-1} (\mathbf{x} - \widehat{\mu}_{\mathbf{x}}) + \widehat{\mu}_y$.

For eleven datasets among a total of fourteen datasets, the NW estimation with the proposed metric statistically achieves one of the best performances. Even when the estimation does not achieve the best performance, the metric always reduces the MSE from the original NW estimation. In particular, in the Slice, Pol, CPU, NavalC, and NavalT datasets, GGL performs poorly showing the non-Gaussianity of data, while the metric using the same information effectively reduces the MSE

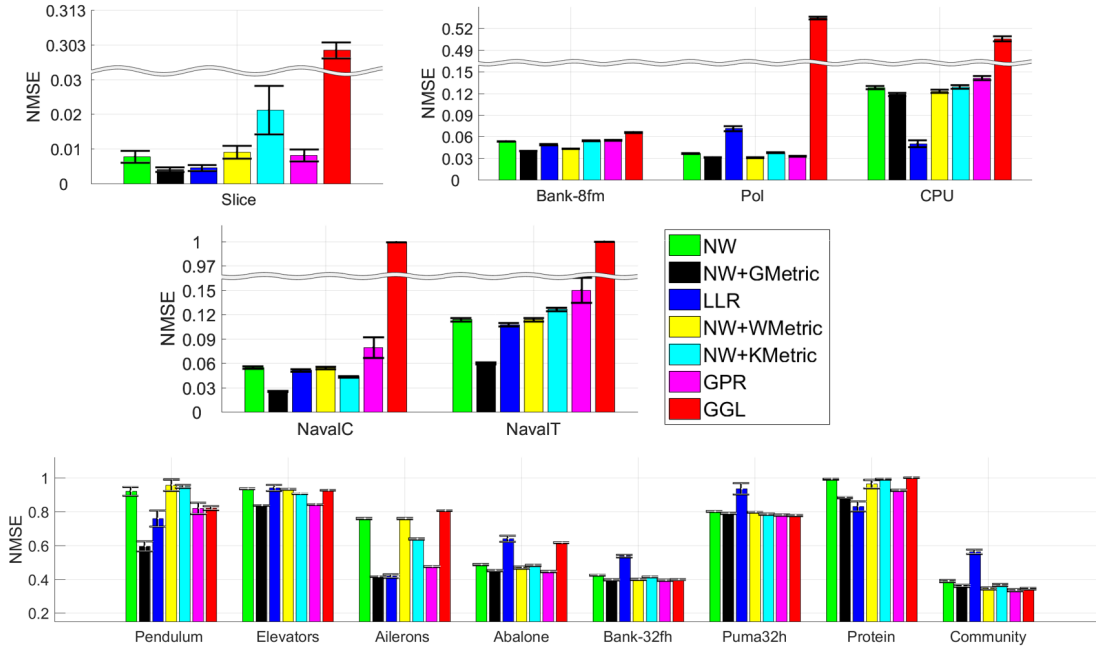

Figure 3: Regression with real-world datasets. NW is the NW regression with conventional kernels, NW+GMetric is the NW regression with the proposed metric, LLR is the locally linear regression, NW+WMetric [28] and NW+KMetric [14] are different metrics for NW regression, GPR is the Gaussian process regression, and GGL is the Gaussian globally linear model. Normalized MSE (NMSE) is the ratio between the MSE and the variance of the target value. If we constantly choose the mean of the target, we get an NMSE of 1.

from the original NW estimator. A detailed discussion comparing the proposed method with other methods for non-Gaussian data is provided in Section 3 and 4 of the Appendix.

# 6  Conclusions

An effective metric function is investigated for reducing the bias of NW regression. Our analysis has shown that the bias can be minimized under certain generative assumptions. The optimal metric is obtained by solving a series of eigenvector problems of size 2 by 2 and needs no explicit gradients or curvature information.

The Gaussian model captures only the rough covariance structure of whole data. The proposed approach uses the *global* covariance to identify the directions that are most likely to have gradient components, and the experiments with real data show that the method is effective for more reliable and less biased estimation. This is in contrast to LLR which attempts to eliminate the linear noise, but the noise elimination relies on a small number of *local* data. In contrast, our model uses additional information from distant data only if they are close in the projected two-dimensional subspace. As a result, the metric allows a more reliable unbiased estimation of the NW estimator.

We have also shown that minimizing the variance is relatively unimportant in high-dimensional spaces compared to minimizing the bias, especially when the bandwidth selection method is used. Consequently, our bias minimization method can achieve sufficiently low MSE without the additional computational cost incurred by empirical MSE minimization.

## Acknowledgments

YKN acknowledges support from NRF/MSIT-2017R1E1A1A03070945, BK21Plus in Korea, MS from KAK-ENHI 17H01760 in Japan, KEK from IITP/MSIT 2017-0-01778 in Korea, FCP from BK21Plus, MITIP-10048320 in Korea, and DDL from the NSF, ONR, ARL, AFOSR, DOT, DARPA in US.

## Footnotes

[1]See Appendix in the supplementary material for the detailed derivation.

[2]See Section 6 of the Appendix:

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
