[Supplementary Material]

# Generative Local Metric Learning for Kernel Regression - Supplementary Material

**Yung-Kyun Noh**
Seoul National University, Rep. of Korea
nohyung@snu.ac.kr

**Masashi Sugiyama**
RIKEN / The University of Tokyo, Japan
sugi@k.u-tokyo.ac.jp

**Kee-Eung Kim**
KAIST, Rep. of Korea
kekim@cs.kaist.ac.kr

**Frank C. Park**
Seoul National University, Rep. of Korea
fcp@snu.ac.kr

**Daniel D. Lee**
University of Pennsylvania, USA
ddlee@seas.upenn.edu

## 1   Metric Dependency of Conditional Expectation, Gradient, and Laplacian

In this section, we show that the conditional expectation of the target variable is metric independent, while the gradients and Laplacians are metric dependent. Due to the independency, our goal becomes clear: by changing metric, we only try to move our estimation toward the conditional expectation, in which case the mean square error (MSE) is always reduced.

Subsection 1.1 contains the proof showing the invariance of the asymptotic estimation result, and the following Subsection 1.2 derives how the gradients and Laplacians change with respect to the change of metric.

### 1.1   Metric dependency of conditional expectation

The conditional expectation $\mathbb{E}[y|\mathbf{x}]$ is invariant to the change of metric at $\mathbf{x}$. If we consider a linear transformation $\mathbf{z} = L^\top \mathbf{x}$ with a full-rank transformation matrix $L$, which is equivalent to a metric change using $A = LL^\top$, it is straightforward to see $\mathbb{E}[y|\mathbf{x}] = \mathbb{E}[y|\mathbf{z}]$ for all $\mathbf{x}$ and their corresponding $\mathbf{z} = L^\top \mathbf{x}$. First, we note that the measure is preserved by the metric change: $p(\mathbf{x})d\mathbf{x} = p(\mathbf{z})d\mathbf{z}$, and $p(y, \mathbf{x})dyd\mathbf{x} = p(y, \mathbf{z})dyd\mathbf{z}$. From these equalities, we get

$$
\begin{aligned}
p(y|\mathbf{x})p(\mathbf{x})dyd\mathbf{x} &= p(y|\mathbf{z})p(\mathbf{z})dyd\mathbf{z} & (1)\\
&= p(y|\mathbf{z})p(\mathbf{x})dyd\mathbf{x} & (2)
\end{aligned}
$$

for all $\mathbf{x}$ and $\mathbf{z}$ and therefore, $p(y|\mathbf{x}) = p(y|\mathbf{z})$ for all corresponding $\mathbf{x}$ and $\mathbf{z}$. Therefore,

$$
\begin{aligned}
\mathbb{E}[y|\mathbf{x}] = \int y\, p(y|\mathbf{x})dy &= \int y\, p(y|\mathbf{z})dy & (3)\\
&= \mathbb{E}[y|\mathbf{z}]. & (4)
\end{aligned}
$$

∎

For example, we consider a joint Gaussian,

$$
p\left(\left(\begin{array}{c}\mathbf{x}\\ y\end{array}\right)\right) = \mathcal{N}\left(\left(\begin{array}{c}\mu_{\mathbf{x}}\\ \mu_y\end{array}\right), \left(\begin{array}{cc}\Sigma_{\mathbf{x}} & \Sigma_{\mathbf{x}y}\\ \Sigma_{y\mathbf{x}} & \Sigma_y\end{array}\right)\right), \tag{5}
$$

and a linear transformation,

$$L' = \begin{pmatrix} L & \vdots \\ & 0 \\ \dots & 0 & 1 \end{pmatrix} \in \mathbb{R}^{(D+1)\times(D+1)}, \tag{6}$$

satisfying $\begin{pmatrix} \mathbf{z} \\ y \end{pmatrix} = L'^\top \begin{pmatrix} \mathbf{x} \\ y \end{pmatrix}$. Then, the mean vector and the covariance matrix in the transformed space are

$$\begin{pmatrix} \mu_{\mathbf{z}} \\ \mu_y \end{pmatrix} = L'^\top \begin{pmatrix} \mu_{\mathbf{x}} \\ \mu_y \end{pmatrix} = \begin{pmatrix} L^\top \mu_{\mathbf{x}} \\ \mu_y \end{pmatrix} \tag{7}$$

and

$$\begin{pmatrix} \Sigma_{\mathbf{z}} & \Sigma_{\mathbf{z}y} \\ \Sigma_{y\mathbf{z}} & \Sigma_y \end{pmatrix} = L'^\top \begin{pmatrix} \Sigma_{\mathbf{x}} & \Sigma_{\mathbf{x}y} \\ \Sigma_{y\mathbf{x}} & \Sigma_y \end{pmatrix} L' \tag{8}$$

$$= \begin{pmatrix} L^\top \Sigma_{\mathbf{x}} L & L^\top \Sigma_{\mathbf{x}y} \\ \Sigma_{y\mathbf{x}} L & \Sigma_y \end{pmatrix}. \tag{9}$$

If we consider the conditional density function in the $\mathbf{x}$-space,

$$y|\mathbf{x} \sim \mathcal{N}(\mu_y + \Sigma_{y\mathbf{x}}\Sigma_{\mathbf{x}}^{-1}(\mathbf{x} - \mu_{\mathbf{x}}), \ \Sigma_y - \Sigma_{y\mathbf{x}}\Sigma_{\mathbf{x}}^{-1}\Sigma_{\mathbf{x}y}), \tag{10}$$

then the conditional density function in the $\mathbf{z}$-space is

$$y|\mathbf{z} \sim \mathcal{N}(\mu_y + \Sigma_{y\mathbf{z}}\Sigma_{\mathbf{z}}^{-1}(\mathbf{z} - \mu_{\mathbf{z}}), \ \Sigma_y - \Sigma_{y\mathbf{z}}\Sigma_{\mathbf{z}}^{-1}\Sigma_{\mathbf{z}y}) \tag{11}$$

$$= \mathcal{N}(\mu_y + \Sigma_{y\mathbf{x}}L(L^\top\Sigma_{\mathbf{x}}L)^{-1}L_{\mathbf{x}}^\top(\mathbf{x} - \mu_{\mathbf{x}}), \tag{12}$$

$$\Sigma_y - \Sigma_{y\mathbf{x}}L(L^\top\Sigma_{\mathbf{x}}L)^{-1}L^\top\Sigma_{\mathbf{x}y}) \tag{13}$$

$$= \mathcal{N}\left(\mu_y + \Sigma_{y\mathbf{x}}\Sigma_{\mathbf{x}}^{-1}(\mathbf{x} - \mu_{\mathbf{x}}), \ \Sigma_y - \Sigma_{y\mathbf{x}}\Sigma_{\mathbf{x}}^{-1}\Sigma_{\mathbf{x}y}\right). \tag{14}$$

This is a re-derivation for a Gaussian that shows $p(y|\mathbf{x})$ and $p(y|\mathbf{z})$ have the same density function, and therefore, $\mathbb{E}[y|\mathbf{x}] = \mathbb{E}[y|\mathbf{z}]$. By the same argument, the conditional variance of $y$, or $\mathbb{E}[\sigma_y^2(\mathbf{x})|\mathbf{x}]$, is invariant to the metric change.

In fact, we note that the equivalence $p(y|\mathbf{x}) = p(y|\mathbf{z})$ can be used without proof from the definition of conditional density function because there is no uncertainty on the transformation of the conditioning variable. In other words, conditioning the random variable $\mathbf{x}$ to a particular value is equivalent to conditioning $\mathbf{z}$ to a transformed fixed value $\mathbf{z} = L^\top\mathbf{x}$, and using the transformed variable $\mathbf{z}$ instead of $\mathbf{x}$ does not affect the density function of $y$.

## 1.2 Metric dependency of gradient and Laplacian

While the conditional density is metric independent, the gradient and Laplacian are metric dependent. In the transformed space using $\mathbf{z} = L^\top\mathbf{x}$, we can show that

$$\nabla_{\mathbf{z}}p(\mathbf{z}) = \frac{1}{|L|}L^{-1}\nabla_{\mathbf{x}}p(\mathbf{x}), \tag{15}$$

and

$$\nabla_{\mathbf{z}}^2 p(\mathbf{z}) = \frac{1}{|L|}tr\left[L^{-1}\nabla_{\mathbf{x}}\nabla_{\mathbf{x}}p \ L^{-\top}\right], \tag{16}$$

where $\nabla_{\mathbf{x}}$ and $\nabla_{\mathbf{z}}$ are the gradients with respect to $\mathbf{x}$ and $\mathbf{z}$, respectively, and $\nabla_{\mathbf{x}}^2$ and $\nabla_{\mathbf{z}}^2$ are the Laplacians with respect to $\mathbf{x}$ and $\mathbf{z}$, respectively. Using $A = LL^\top$,

$$\nabla_{\mathbf{z}}^\top p(\mathbf{z})\nabla_{\mathbf{z}}p(\mathbf{z}) = \frac{1}{|A|}tr[A^{-1}\nabla_{\mathbf{x}}p\nabla_{\mathbf{x}}^\top p], \tag{17}$$

$$\nabla_{\mathbf{z}}^2 p(\mathbf{z}) = \frac{1}{|A|^{\frac{1}{2}}}tr\left[A^{-1}\nabla_{\mathbf{x}}\nabla_{\mathbf{x}}p\right]. \tag{18}$$

The metric dependency provides an optimization problem for the minimization of the bias and MSE.

For the derivative of general functions, the coordinate transformation satisfies the relationship:

$$\nabla_{\mathbf{z}}f(\mathbf{z}) = L^{-1}\nabla_{\mathbf{x}}f(\mathbf{x}). \tag{19}$$

### 1.2.1 Detailed derivation

The detailed derivations of Eq. (15) and Eq. (16) are as follows:

We consider a linear transformation $\mathbf{z} = L^\top \mathbf{x}$ and the conservation of measure $p(\mathbf{x})d\mathbf{x} = p(\mathbf{z})d\mathbf{z}$. Then we can write

$$
\begin{aligned}
p(\mathbf{z})d\mathbf{z} &= p(\mathbf{z} = L^\top \mathbf{x})|L|d\mathbf{x} & (20)\\
&= p(\mathbf{x})d\mathbf{x}. & (21)
\end{aligned}
$$

Therefore, $p(\mathbf{x}) = |L|p(\mathbf{z} = L^\top \mathbf{x})$.

Now, the $i$-th element of the gradient of density function is

$$
\begin{aligned}
(\nabla_{\mathbf{z}} p(\mathbf{z}))_i &= \frac{\partial p(\mathbf{z})}{\partial z_i} = \frac{1}{|L|} \sum_j \frac{\partial p(\mathbf{x})}{\partial x_j} \frac{\partial x_j}{\partial z_i} & (22)\\
&= \frac{1}{|L|} \sum_j \frac{\partial p(\mathbf{x})}{\partial x_j} (L^{-\top})_{ji} & (23)\\
&= \frac{1}{|L|} (L^{-1} \nabla p(\mathbf{x}))_i, & (24)
\end{aligned}
$$

and for the Laplacian, the second derivative with respect to the $i$-th element is

$$
\begin{aligned}
\frac{\partial^2 p(\mathbf{z})}{\partial z_i^2} &= \frac{1}{|L|} \frac{\partial}{\partial z_i} (L^{-1} \nabla p(\mathbf{x}))_i & (25)\\
&= \frac{1}{|L|} \sum_k \frac{\partial}{\partial x_k} (L^{-1} \nabla p(\mathbf{x}))_i \frac{\partial x_k}{\partial z_i} & (26)\\
&= \frac{1}{|L|} \sum_k \sum_j \frac{\partial}{\partial x_k} (L^{-1})_{ij} \frac{\partial p(\mathbf{x})}{\partial x_j} \frac{\partial x_k}{\partial z_i} & (27)\\
&= \frac{1}{|L|} \sum_k \sum_j (L^{-1})_{ij} \frac{\partial^2 p(\mathbf{x})}{\partial x_k \partial x_j} (L^{-\top})_{ki} & (28)\\
&= \frac{1}{|L|} (L^{-1} \nabla \nabla p(\mathbf{x}) L^{-\top})_{ii}. & (29)
\end{aligned}
$$

Therefore, the Laplacian of the density function is

$$
\nabla_{\mathbf{z}}^2 p(\mathbf{z}) = \sum_i \frac{\partial^2 p(\mathbf{z})}{\partial z_i^2} = \frac{1}{|L|} tr \left[ L^{-1} \nabla \nabla p(\mathbf{x}) L^{-\top} \right]. \tag{30}
$$

The derivative of general function $f(\mathbf{x})$ is different from the derivative of the density functions:

$$
\begin{aligned}
(\nabla_{\mathbf{z}} f)_i &= \frac{\partial f}{\partial z_i} = \sum_j \frac{\partial f}{\partial x_j} \frac{dx_j}{dz_i} & (31)\\
&= \sum_j \frac{\partial f}{\partial x_j} (L^{-1})_{ij} & (32)\\
&= \left[ L^{-1} \frac{\partial f}{\partial \mathbf{x}} \right]_i. & (33)
\end{aligned}
$$

## 2  Comparison with Locally Linear Regression and Gaussian Processes

The weakness of the Nadaraya-Watson (NW) estimator is explained in Chapter 6 of [4], where the boundary issue in a low-dimensional space is the main motivation for extending the NW estimator to Locally Linear Regression (LLR) [1, 17].

LLR introduces two parameter variables $\alpha \in \mathbb{R}$ and $\beta \in \mathbb{R}^D$ and finds the parameters by solving the LLR optimization problem with training data $\mathcal{D} = \{\mathbf{x}_i, y_i\}_{i=1}^N$:

$$\alpha(\mathbf{x}), \beta(\mathbf{x}) = \arg \min_{\alpha \in \mathbb{R}, \beta \in \mathbb{R}^D} \sum_{i=1}^N \left(y_i - \alpha - \beta^\top (\mathbf{x}_i - \mathbf{x})\right)^2 K(\mathbf{x}_i, \mathbf{x}) \tag{34}$$

where $\alpha$ is the estimation of our target value $y(\mathbf{x})$, and $\beta$ is a nuisance parameter with the estimation of $\nabla y$. With the kernel function $K(\mathbf{x}_i, \mathbf{x})$, nearby data contribute more for the estimation of $\alpha$ and $\beta$, otherwise the model is a globally linear model without kernel. With simple calculation, the solution can be obtained in a closed form:

$$\begin{pmatrix} \alpha \\ \beta \end{pmatrix} = (X_\mathbf{x} W_\mathbf{x} X_\mathbf{x}^\top)^{-1} X_\mathbf{x} W_\mathbf{x} \mathbf{y} \tag{35}$$

with the following matrices:

$$X_\mathbf{x} = \begin{pmatrix} 1 & (\mathbf{x}_1 - \mathbf{x})^\top \\ \vdots & \vdots \\ 1 & (\mathbf{x}_N - \mathbf{x})^\top \end{pmatrix}^\top, \tag{36}$$

$$W_\mathbf{x} = diag\{K(\mathbf{x}_1, \mathbf{x}), \ldots, K(\mathbf{x}_N, \mathbf{x})\}, \text{ and} \tag{37}$$

$$\mathbf{y} = [y_1, \ldots, y_N]^\top. \tag{38}$$

Why is LLR the extension of the NW estimator? If we consider a simpler objective function

$$\sum_{i=1}^N (y_i - \alpha)^2 K(\mathbf{x}_i, \mathbf{x}), \tag{39}$$

we can find the optimal $a$ minimizing the objective function with a closed form solution:

$$a = \frac{\sum_i K(\mathbf{x}_i, \mathbf{x}) y_i}{\sum_i K(\mathbf{x}_i, \mathbf{x})}, \tag{40}$$

which is the solution of NW estimation. Therefore, we can consider LLR as the extension of the NW estimator which estimates the slope of $y(\mathbf{x})$ as well as $y(\mathbf{x})$ at the point of interest $\mathbf{x}$. In fact, if the vector $\beta\, (= \nabla y(\mathbf{x}))$ in LLR is estimated correctly, it eliminates the bias of $\alpha$ due to the slope of $y(\mathbf{x})$ and asymptotically, the bias of LLR is

$$\text{Bias}_{\text{LLR}} = \frac{h^2 \nabla^2 y(\mathbf{x})}{2} \tag{41}$$

where the asymptotic bias can be compared with that of the NW estimator:

$$\text{Bias}_{\text{NW}} = h^2 \left( \frac{\nabla^\top p(\mathbf{x}) \nabla y(\mathbf{x})}{p(\mathbf{x})} + \frac{\nabla^2 y(\mathbf{x})}{2} \right) \tag{42}$$

with an additional term $h^2 \frac{\nabla^\top p(\mathbf{x}) \nabla y(\mathbf{x})}{p(\mathbf{x})}$ with the slope of $y(\mathbf{x})$, $\nabla y(\mathbf{x})$.

Note that the asymptotic variances for the two estimators are the same:

$$\text{Var}_{\text{LLR}} = \text{Var}_{\text{NW}} = \frac{\sigma_y^2(\mathbf{x})}{N h^D (2\sqrt{\pi})^D p(\mathbf{x})}. \tag{43}$$

Asymptotically, the additional fitting of the slope $y(\mathbf{x})$ with more parameters eliminates the bias due to that slope. However in many experiments with real data, it is reported that LLR does not necessarily outperform NW estimation [2, 5]. Similarly, our experiments incurred high dimensional properties by using more parameters, and LLR was prone to overfit.

## 2.1 Locally Linear Regression and Our Metric Learning

Interestingly, both LLR and our method tackle the first term of the asymptotic bias, $h^2 \frac{\nabla^\top p(\mathbf{x}) \nabla y(\mathbf{x})}{p(\mathbf{x})}$. Our metric eliminates this term by choosing a particular metric $A$, and LLR does so by adopting more

parameters $\beta$. When the true $y(\mathbf{x})$ is a linear function with $\nabla^2 y(\mathbf{x}) = 0$, both methods eliminate the asymptotic bias.

However, the two methods are completely different, and the experiments with linear and nonlinear $y(\mathbf{x})$ show that choosing a metric gives better results than LLR. In the ideal situation where $y$ and $\mathbf{x}$ are jointly Gaussian, LLR performs very well, as depicted in Fig. 1(a) when the linearity assumption in LLR is satisfied, but once the linearity assumption breaks, the NW estimator usually performs better than LLR, as shown in Fig. 1(b). In Fig. 1(b) when the Gaussian assumption is broken, it is obvious that LLR does not make a good prediction. LLR is sensitive to the break of the assumption, while the proposed metric simply reduces the MSE even with non-Gaussian data.

(a) $y(\mathbf{x})$ is linear          (b) $y(\mathbf{x})$ is quadratic

Figure 1: LLR in high dimensional space (a) Linear condition is satisfied in 30-dimensional space, with number of data 300. (b) Target function is quadratic in 50-dimensional space, with number of data 500.

Our metric uses the estimated Gaussian parameters, but in Fig. 1(a), we also present the MSE using the metric from the "true parameters" (red). From the red curve, we can see that once we achieve the true gradients, the result is better than the best result of LLR with optimal bandwidth. We also note that in practice, the regime of bandwidth achieving the minimum MSE is very narrow for LLR (light blue), and often the chosen bandwidth from training data does not perform well for testing data. On the other hand, according to Fig. 1(a), the small MSE is achieved in metric learning throughout a wide regime of bandwidth (black), and even a rough choice of bandwidth does not increase the MSE significantly.

If we summarize the comparison between NW estimation with metric and LLR,

- Both use completely different approaches for regression, but they show the same asymptotic results; they eliminate the first term of the bias, $h^2 \frac{\nabla^\top p(\mathbf{x}) \nabla y(\mathbf{x})}{p(\mathbf{x})}$.

- NW estimation obtains the metric information from a generative model (one Gaussian) using all data globally, while LLR uses only local data to estimate both the $y(\mathbf{x})$ and $\nabla y(\mathbf{x})$.

- Empirically, LLR can achieve a very small MSE but only with an exactly chosen bandwidth. The result is sensitively corrupted by the perturbation of bandwidth. While the MSE from metric learning is insensitive to the choice of bandwidth.

## 2.2 Gaussian Process Regression and Locally Linear Regression

Although NW estimation, LLR, and Gaussian process regression (GPR) are the three most well-known methods in nonparametric regression, the relationship among them has not been extensively investigated. GPR considers an infinite dimensional Gaussian, and the regression performs the inference using the mean of the conditional density function:

$$\widehat{y}(\mathbf{x}) = \mathbf{k}^\top K^{-1} \mathbf{y} \tag{44}$$

where $\mathbf{k}$ is the vector with $i$-th element $\mathbf{k}_i = K(\mathbf{x}_i, \mathbf{x})$, $K$ is the matrix with $K_{ij} = K(\mathbf{x}_i, \mathbf{x}_j)$, and $\mathbf{y}$ is the vector with its element $\mathbf{y}_i = y_i$. In terms of computational complexity, GPR needs to

calculate the inverse of the $N \times N$ matrix $K$ with the number of data $N$, and LLR needs the inverse of $(D+1) \times (D+1)$ matrix $X_{\mathbf{x}} W_{\mathbf{x}} X_{\mathbf{x}}^{\top}$ with the dimensionality $D$.

First, we note that we can prove LLR is a GPR with a particular choice of GPR kernel. Using the inverse identity $(P^{-1} + B^{\top} R^{-1} B)^{-1} B^{\top} R^{-1} = P B^{\top} (B P B^{\top} + R)^{-1}$ with $R = W_{\mathbf{x}}^{-1}, B = X_{\mathbf{x}}, P = \frac{1}{\gamma} I$ with small $\gamma$, the following $(D+1) \times N$ matrix can be reformulated

$$
\begin{aligned}
(X_{\mathbf{x}}^{\top} W_{\mathbf{x}} X_{\mathbf{x}})^{-1} X_{\mathbf{x}}^{\top} W_{\mathbf{x}} &= \lim_{\gamma \to 0} (X_{\mathbf{x}}^{\top} W_{\mathbf{x}} X_{\mathbf{x}} + \gamma I)^{-1} X_{\mathbf{x}}^{\top} W_{\mathbf{x}} \quad (45)\\
&= \lim_{\gamma \to 0} \frac{1}{\gamma} I X_{\mathbf{x}}^{\top} (X_{\mathbf{x}} \frac{1}{\gamma} I X_{\mathbf{x}}^{\top} + W^{-1})^{-1} \quad (46)\\
&= \lim_{\gamma \to 0} X_{\mathbf{x}}^{\top} (X_{\mathbf{x}} X_{\mathbf{x}}^{\top} + \gamma W^{-1})^{-1}. \quad (47)
\end{aligned}
$$

Here, $L \equiv X_{\mathbf{x}} X_{\mathbf{x}}^{\top} + \gamma W^{-1} \in \mathbb{R}^{N \times N}$ is a new kernel matrix with element

$$
L_{ij} = \begin{pmatrix} 1 \\ \mathbf{x}_i - \mathbf{x} \end{pmatrix}^{\top} \begin{pmatrix} 1 \\ \mathbf{x}_j - \mathbf{x} \end{pmatrix} + \frac{\gamma}{\sqrt{K(\mathbf{x}_i, \mathbf{x}) K(\mathbf{x}_j, \mathbf{x})}} \delta_{ij}. \quad (48)
$$

If we define a column vector $\mathbf{l} \in \mathbb{R}^N$ with a kernel element between $\mathbf{x}_i$ and $\mathbf{x}$: $\mathbf{l}_i = \begin{pmatrix} 1 \\ \mathbf{x} - \mathbf{x} \end{pmatrix}^{\top} \begin{pmatrix} 1 \\ \mathbf{x}_i - \mathbf{x} \end{pmatrix} = e_1^{\top} \begin{pmatrix} 1 \\ \mathbf{x}_i - \mathbf{x} \end{pmatrix}$ with the column vector $e_1 = [1 \ 0 \ \dots \ 0]^{\top}$.

Now we consider the regression function using an LLR closed form solution and reformulate it:

$$
\begin{aligned}
\widehat{y}(\mathbf{x}) &= e_1^{\top} (X_{\mathbf{x}}^{\top} W_{\mathbf{x}} X_{\mathbf{x}})^{-1} X_{\mathbf{x}}^{\top} W_{\mathbf{x}} \mathbf{y} \quad (49)\\
&= e_1^{\top} X_{\mathbf{x}}^{\top} (X_{\mathbf{x}} X_{\mathbf{x}}^{\top} + \gamma W^{-1})^{-1} \mathbf{y} \quad (50)\\
&= \mathbf{l}^{\top} L^{-1} \mathbf{y} \quad (51)
\end{aligned}
$$

which is the same formulation as Eq. (44). ∎

Now we can understand LLR as a special problem of using the kernel defined in Eq. (48) and consider the GPR as an extending algorithm of the NW estimator. Without any intervening metric, LLR and GPR have each proposed their own ways of alleviating the bias for nonparametric regression with kernels. The major difference between NW estimation and LLR/GPR is that the NW estimator only performs interpolation. The prediction never gives smaller or greater values than the minimum or maximum of the training outputs, respectively. In particular near the boundary, the unbalanced data inside and outside the boundary produces bias toward the target values of data inside the boundary.

LLR and GPR do not produce this boundary issue using the extrapolation. LLR uses the slope estimates using data inside the boundary [4], and it extrapolates using the estimated slope information. GPR uses an even more flexible method for extrapolation.

In high dimensional space, flexible models suffer from the sparsity of data. For example, LLR estimates the target value as well as its slope to help reduce the bias, but the slope estimation is prone to overfit in high dimensional space. Without an appropriate choice of metric, the bias reduction of flexible models is limited.

## 3 Local vs. Global Information

Algorithms can use local or global information. Nonparametric methods tend to use *local* information from nearby data. Algorithms using global information use the statistics from all data. Total mean and total covariance are the global information, and globally linear models also use global information.

Because global information cannot capture the detailed shape of the target function even with many data, a global model cannot perform better than the local model when many data are given. However in high dimensional space, the number of data cannot be large enough, so often global algorithms work more robustly than the local model by using effectively more data. The discussion is similar to that between discriminative and generative models [11].

The NW estimator is purely a local algorithm; the information from the distant data does not affect the result. Global configuration of data is never used for obtaining target value, and once significant noise

is applied to nearby data, the result becomes poor. Our metric learning uses a global configuration of data, capturing the rough covariance structure of whole data and recommends the NW estimator to give more attention to to the direction and importance of data. Without this metric, the NW estimator simply ignores the information from global configuration.

LLR and GPR use only local information for prediction. In Table 1, we summarize the information used by different algorithms. In the table, GMetric is the Gaussian metric in our paper. A global metric was introduced to minimize the empirical MSE [21] (WMetric), and the metric introduced in [8, 9] also uses a global configuration of data (KMetric). In KMetric, the average gradient of each coordinate is estimated to diminish the weight along the direction having a large average gradient. Naturally, NW, LLR, and GPR themselves are vulnerable to noise because the effective amount of utilized data is relatively small. However, the three metrics using *global* configuration of data are robust to a small perturbation by noise.

Lastly, our work uses a metric that is different at each point, in contrast to WMetric and KMetric which use a single metric throughout the space. Our metric depends on $\nabla p(\mathbf{x})$ at each point $\mathbf{x} \in \mathbb{R}^D$, but the gradient information comes from the global configuration of data which is robust to noise. Although our metric uses *global* information, the applied metric is *local* depending on the point of interest. Therefore, the proposed metric is the *local* metric using *global* information.

### 3.1 Non-Gaussianity of Data

The actual prediction is performed by the NW estimator, and the chosen metric only provides the guidelines for more attention (kernel weights). Because the metric does not directly perform prediction but only helps NW prediction, the non-Gaussianity of the raw-data is not critically harmful. Once we have knowledge of all true gradients and Laplacians of data, we do not need a Gaussian model, but a metric can be directly obtained with amazing NW estimation accuracy, as shown in Fig.2(b) in the main manuscript.

If the metric is obtained from the information vulnerable to noise, it can be critically harmful. The equation for the metric has to be simple and robust. By adopting a single Gaussian, we can obtain a very simple, robust, and quick-to-calculate equation for metric from the analytic properties of Gaussian.

In what follows, we summarize the reasons why the proposed algorithm provides empirically good results for all eleven regression datasets:

- The NW estimator is a local algorithm which is very flexible but needs extremely many data. The algorithm is supported by a metric learned from global information.

- A Gaussian model could provide an analytically derived metric. The metric equation is simple and robust to noise.

- The proposed metric is a local metric that is more flexible than the global metrics such as WMetric and KMetric. Though the metric does not change through simple perturbation of data by using global configuration of data.

- Non-Gaussianity does not necessarily make harmful results. Our rough Gaussian model only captures the Gaussian component of the underlying density function with minimal KL-divergence between the chosen Gaussian and the underlying density function.

- The obtained two-dimensional metric has the effect of dimensionality reduction. We are now able to use the information from data orthogonal to the two-dimensional space, and the proposed metric effectively lets the estimator use more data.

## 4 Comparison with Generative Local Metric for Nearest Neighbor Classification

An idea for using rough generative models for metric learning is previously proposed in [13] for nearest neighbor (NN) classification, and the research is extended to [14] providing competitive results with the state of the art algorithms. Similar ideas have also appeared using asymptotic MSE for NN classification, though this research does not use generative models [3]. In related research, the asymptotic bias of NN classification is derived but not used for metric learning [20].

Table 1: Comparison of information for different algorithms

| ALGORITHM | INFORMATION FOR TRAIN | OBTAINED METRIC |
|---|---|---|
| NW | LOCAL | |
| GMETRIC FOR NW | GLOBAL | LOCAL |
| LLR | LOCAL | |
| GPR | LOCAL | |
| WMETRIC FOR NW | GLOBAL | GLOBAL |
| KMETRIC FOR NW | GLOBAL | GLOBAL |

NW: NADARAYA-WATSON ESTIMATOR
GMETRIC: GAUSSIAN METRIC
LLR: LOCALLY LINEAR REGRESSION
GPR: GAUSSIAN PROCESS REGRESSION
WMETRIC: WEINBERGER'S METRIC [21]
KMETRIC: KPOTUFE'S METRIC [8, 9]

The proposed method is closely related to [13]. The idea is to use one Gaussian generative model to capture the rough covariance structure and help a nonparametric algorithm reduce the bias. The result of capturing the rough structure and applying the analytically derived metric efficiently reduces the error for various datasets which are not necessarily close to Gaussian [14].

We summarize the advances from these previous studies [13, 14], and they are as follows:

- [13, 14] are designed for NN classification, and our metric is derived for NW estimation.

- Our metric for NW regression has a nice two-dimensional structure, while [13, 14] do not.

- The guarantee of the bias elimination for Gaussian is only applied to our method.

- Variance analysis is provided in our discussion, which also can possibly be applied back to [13, 14].

- The metric depends on the gradient information which is more robustly estimated from data than Laplacians, which are the main information for the metric in [13, 14].

## 5 Exemplar Models for Decision Making

In cognitive science, the NW estimator formulation is the most often used. Since the introduction of Shepard to the exponential law [18], kernel regression formulation in NW estimation has become the standard formulation for explaining human decision making [6, 19]. The set of models using NW estimation are called the exemplar model for describing the psychological distance of human memory.

The mathematical knowledge on distances and kernels can influence the cognitive modeling [12, 7]. It is also shown in [15] that exemplar model formulation is closely connected to another famous drift diffusion model for decision making [16]. However, none of these methods has attempted to treat the metric for better performance of decision making as far as the authors know.

## 6 Derivation of the Bias and the Variance of Nadaraya-Watson Estimation

### 6.1 Bias of Nadaraya-Watson regression

This section presents a detailed derivation of the Nadaraya-Watson (NW) bias. The derivation here is a re-derivation of the bias from several previous studies [10, 17], though none was ever published using the derived bias equation for metric learning. The bias we derive is the expected deviation of the estimator from the true mean of the target variable $y(\mathbf{x})$:

$$\text{Bias} = \mathbb{E}\left[\widehat{y}(\mathbf{x}) - y(\mathbf{x})\right] \tag{52}$$

$$= \mathbb{E}\left[\frac{\sum_{i=1}^{N} K(\mathbf{x}_i, \mathbf{x})y_i}{\sum_{i=1}^{N} K(\mathbf{x}_i, \mathbf{x})} - y(\mathbf{x})\right]. \tag{53}$$

Assuming concentration in the denominator and the numerator, we obtain the bias. First, for the denominator,

$$\mathbb{E}_{\mathbf{x}_1,\ldots,\mathbf{x}_N}\left[\frac{1}{N}\sum_{i=1}^{N}K(\mathbf{x}_i,\mathbf{x})\right] = \mathbb{E}_{\mathbf{t}}\left[K(\mathbf{t},\mathbf{x})\right] \tag{54}$$

$$= \int p(\mathbf{t})K\left(\frac{\mathbf{t}-\mathbf{x}}{h}\right)d\mathbf{t} \tag{55}$$

$$= \int \left(p(\mathbf{x}+h\mathbf{z})\right)K(\mathbf{z})d\mathbf{z} \tag{56}$$

$$= \int \left(p(\mathbf{x})+h\mathbf{z}^{\top}\nabla p+\frac{h^2}{2}\mathbf{z}^{\top}\nabla\nabla p\mathbf{z}+\mathcal{O}(h^3))\right)K(\mathbf{z})d\mathbf{z} \tag{57}$$

$$= p(\mathbf{x})+\frac{h^2}{2}\nabla^2 p+\mathcal{O}(h^4), \tag{58}$$

where we used the change of variable $\frac{\mathbf{t}-\mathbf{x}}{h}=\mathbf{z}$ to obtain Eq. (56) from Eq. (55). The Jacobian relationship, $d\mathbf{t}=h^D d\mathbf{z}$ with dimensionality $D$, is used, and $h^D$ from the Jacobian relationship is canceled by the change in kernel: $K\left(\frac{\mathbf{t}-\mathbf{x}}{h}\right)=K(\mathbf{z})/h^D$. For the integration in Eq. (57) leading to Eq. (58), we assume that we are using a kernel satisfying $\int K(\mathbf{z})d\mathbf{z}=1$, $\int \mathbf{z}K(\mathbf{z})d\mathbf{z}=0$, and $\int \mathbf{z}\mathbf{z}^{\top}K(\mathbf{z})d\mathbf{z}=I$ (e.g. a Gaussian kernel). Similarly, the expectation of the numerator becomes

$$\mathbb{E}_{\mathbf{x}_1,\ldots,\mathbf{x}_N,y_1,\ldots,y_N}\left[\frac{1}{N}\sum_{i=1}^{N}K(\mathbf{x},\mathbf{x}_i)y_i\right] = \mathbb{E}_{\mathbf{t},y}\left[K(\mathbf{t},\mathbf{x})y\right] \tag{59}$$

$$= \int p(y)y\cdot p(\mathbf{t}|y)K\left(\frac{\mathbf{t}-\mathbf{x}}{h}\right)d\mathbf{t}dy \tag{60}$$

$$= \int p(y)y\cdot\left(p(\mathbf{x}|y)+\frac{h^2}{2}\nabla^2 p(\mathbf{x}|y)+\mathcal{O}(h^4)\right)dy \tag{61}$$

$$= \int p(\mathbf{x},y)ydy+\frac{h^2}{2}\int y\nabla^2 p(\mathbf{x},y)dy+\mathcal{O}(h^4) \tag{62}$$

$$= p(\mathbf{x})y(\mathbf{x})+\frac{h^2}{2}\int y\nabla^2 p(\mathbf{x},y)dy+\mathcal{O}(h^4). \tag{63}$$

In the second order term of $h$, we can further calculate $\int y\nabla^2 p(\mathbf{x},y)dy = \nabla^2 \int y\cdot p(\mathbf{x},y)dy = 2\nabla p(\mathbf{x})^{\top}\nabla y(\mathbf{x})+y(\mathbf{x})\nabla^2 p+p(\mathbf{x})\nabla^2 y(\mathbf{x})$, and the bias can be approximated up to the second order of $h$:

$$\mathbb{E}\left[\frac{\sum_{i=1}^{N}K(\mathbf{x},\mathbf{x}_i)y_i}{\sum_{i=1}^{N}K(\mathbf{x},\mathbf{x}_i)}-y(\mathbf{x})\right] = h^2\left(\frac{\nabla^{\top}p(\mathbf{x})\nabla y(\mathbf{x})}{p(\mathbf{x})}+\frac{\nabla^2 y(\mathbf{x})}{2}\right)+\mathcal{O}(h^4). \tag{64}$$

## 6.2 Variance of Nadaraya-Watson regression

The variance of the NW estimator is known to be $\frac{1}{Nh^D(2\sqrt{\pi})^D}\frac{\sigma_y^2(\mathbf{x})}{p(\mathbf{x})}$ with conditional variance of $y$, $\sigma_y^2(\mathbf{x})=\mathbb{E}\left[y^2|\mathbf{x}\right]-\mathbb{E}\left[y|\mathbf{x}\right]^2=\int y^2 p(y|\mathbf{x})dy-y(\mathbf{x})^2$ [4]. In fact, the asymptotic variance up to the order of $h^{2-D}$ can be obtained as well, and we provide the derivation of the variance up to the order of $h^{2-D}$ in this section.

First, we consider the MSE of the NW estimator:

$$\mathbb{E}\left[\left(\frac{\sum_{i=1}^{N}K(\mathbf{x}_i,\mathbf{x})y_i}{\sum_{i=1}^{N}K(\mathbf{x}_i,\mathbf{x})}-y(\mathbf{x})\right)^2\right] = \mathbb{E}\left[\frac{\left(\sum_{i=1}^{N}K(\mathbf{x}_i,\mathbf{x})y_i-y(\mathbf{x})\sum_{i=1}^{N}K(\mathbf{x}_i,\mathbf{x})\right)^2}{\left(\sum_{i=1}^{N}K(\mathbf{x}_i,\mathbf{x})\right)^2}\right]. \tag{65}$$

For approximation with perturbation, we calculate the expectation of the nominator and denominator separately. The denominator can be approximated with

$$\mathbb{E}\left[\left(\frac{1}{N}\sum_{i=1}^{N}K(\mathbf{x}_i,\mathbf{x})\right)^2\right] = N\mathbb{E}\left[K(\mathbf{t},\mathbf{x})^2\right]+\left(1-\frac{1}{N}\right)\mathbb{E}\left[K(\mathbf{t},\mathbf{x})\right]^2, \tag{66}$$

and the nominator can be approximated with

$$\mathbb{E}\left[\left(\frac{1}{N}\sum_{i=1}^{N}K(\mathbf{x}_i,\mathbf{x})y_i - y(\mathbf{x})\frac{1}{N}\sum_{i=1}^{N}K(\mathbf{x}_i,\mathbf{x})\right)^2\right] \tag{67}$$

$$= \mathbb{E}\left[\left(\frac{1}{N}\sum_{i=1}^{N}K(\mathbf{x}_i,\mathbf{x})y_i\right)^2\right] + y(\mathbf{x})^2\mathbb{E}\left[\left(\frac{1}{N}\sum_{i=1}^{N}K(\mathbf{x}_i,\mathbf{x})\right)^2\right] \tag{68}$$

$$- 2y(\mathbf{x})\mathbb{E}\left[\frac{1}{N^2}\left(\sum_{i=1}^{N}K(\mathbf{x}_i,\mathbf{x})\sum_{i=1}^{N}K(\mathbf{x}_i,\mathbf{x})y_i\right)\right]$$

$$= \frac{1}{N}\mathbb{E}\left[K(\mathbf{t},\mathbf{x})^2y^2\right] + \left(1+\frac{1}{N}\right)\mathbb{E}\left[K(\mathbf{t},\mathbf{x})y\right]^2 \tag{69}$$

$$+ \frac{y(\mathbf{x})^2}{N}\mathbb{E}\left[K(\mathbf{t},\mathbf{x})^2\right] + y(\mathbf{x})^2\left(1+\frac{1}{N}\right)\mathbb{E}\left[K(\mathbf{t},\mathbf{x})\right]^2$$

$$- 2\frac{y(\mathbf{x})}{N}\mathbb{E}\left[K(\mathbf{t},\mathbf{x})^2y\right] - 2y(\mathbf{x})\left(1+\frac{1}{N}\right)\mathbb{E}\left[K(\mathbf{t},\mathbf{x})\right]\mathbb{E}\left[K(\mathbf{t},\mathbf{x})y\right].$$

With similar derivation with the bias, the expectation can be calculated as

$$\mathbb{E}\left[K(\mathbf{t},\mathbf{x})^2y^2\right] = \frac{1}{h^D}\frac{1}{(2\sqrt{\pi})^D}\left\{p(\mathbf{x})(\sigma_y^2 + y(\mathbf{x})^2) + \frac{h^2}{4}\nabla^2(\sigma_y^2 + y(\mathbf{x})^2)p(\mathbf{x}) + \mathcal{O}(h^4)\right\}, \tag{70}$$

$$\mathbb{E}\left[K(\mathbf{t},\mathbf{x})^2y\right] = \frac{1}{h^D}\frac{1}{(2\sqrt{\pi})^D}\left\{p(\mathbf{x})y(\mathbf{x}) + \frac{h^2}{4}\nabla^2[y(\mathbf{x})p(\mathbf{x})] + \mathcal{O}(h^4)\right\}, \tag{71}$$

$$\mathbb{E}\left[K(\mathbf{t},\mathbf{x})^2\right] = \frac{1}{h^D}\frac{1}{(2\sqrt{\pi})^D}\left\{p(\mathbf{x}) + \frac{h^2}{4}\nabla^2 p + \mathcal{O}(h^4)\right\}. \tag{72}$$

By plugging the above three equations, Eq. (70), (71), (72) and the expectations in the previous section, Eq. (58), (63) into Eq. (69), we can obtain the following approximation of the MSE:

$$\mathbb{E}\left[\left(\frac{\sum_{i=1}^{N}K(\mathbf{x}_i,\mathbf{x})y_i}{\sum_{i=1}^{N}K(\mathbf{x}_i,\mathbf{x})} - y(\mathbf{x})\right)^2\right] \tag{73}$$

$$\approx \underbrace{h^4\left(\frac{\nabla^2 y(\mathbf{x})}{2} + \frac{\nabla^\top y(\mathbf{x})\nabla p(\mathbf{x})}{p(\mathbf{x})}\right)^2}_{Bias^2} \tag{74}$$

$$+ \underbrace{\frac{1}{Nh^D(2\sqrt{\pi})^D}\left\{\frac{\sigma_y^2(\mathbf{x})}{p(\mathbf{x})} + \frac{h^2}{2}\left(\frac{\nabla^2[\sigma_y^2(\mathbf{x})p(\mathbf{x})]}{2p(\mathbf{x})^2} + \frac{(\nabla y(\mathbf{x}))^2}{p(\mathbf{x})}\right) + \mathcal{O}(h^4)\right\}}_{Variance} \tag{75}$$

With the above derivation, the variance of the NW estimator can be derived as follows:

$$\text{Variance} = \mathbb{E}[(\widehat{y}(\mathbf{x}) - \mathbb{E}[\widehat{y}(\mathbf{x})])^2] \tag{76}$$

$$= \frac{1}{Nh^D(2\sqrt{\pi})^D}\left[\frac{\sigma_y^2(\mathbf{x})}{p(\mathbf{x})} + \frac{h^2}{2}\left(\frac{\nabla^2[\sigma_y^2(\mathbf{x})p(\mathbf{x})]}{2p(\mathbf{x})^2} + \frac{(\nabla y(x))^2}{p(\mathbf{x})}\right) + \mathcal{O}(h^4)\right], \tag{77}$$

with $\sigma_y^2(\mathbf{x}) = \int y^2 p(y|\mathbf{x})dy - y(\mathbf{x})^2$ which is the conditional variance of $y$. For Gaussian data, $\sigma_y^2(\mathbf{x})$ is a constant, and the variance can be modified as

$$\text{Variance}_{Gaussian} = \frac{1}{Nh^D(2\sqrt{\pi})^D}\left[\frac{\sigma_y^2}{p(\mathbf{x})} + \frac{h^2}{2}\left(\frac{\sigma_y^2}{2\,p(\mathbf{x})^2}\nabla^2 p + \frac{(\nabla y(x))^2}{p(\mathbf{x})}\right) + \mathcal{O}(h^4)\right]. \tag{78}$$

(a)

Figure 2: Empirical MSE of the NW estimator with generated data and calculated bias$^2$ and MSE. Var$_1$ is the leading term of the derived variance of order $h^{-D}$. Var$_2$ is the variance up to the order of $h^{2-D}$. Data are generated using 3-dimensional Gaussian data where one dimensionality is used for $y$ and the other two dimensionalities are used for $\mathbf{x}$.

In a low-dimensional space, the MSE in Eq. (74) and Eq. (75) approximate the empirical MSE as shown in Fig.2. The leading term of the variance is $\frac{1}{Nh^D(2\sqrt{\pi})^D}\frac{\sigma_y^2(\mathbf{x})}{p(\mathbf{x})}$, and this term is mainly caused by the noise upon the target value $y$, and $\sigma_y^2(\mathbf{x})$ is invariant to the metric for $\mathbf{x}$. We explained in Section 4.3 of the main manuscript how we can effectively reduce the MSE with this leading term. Instead of trying to reduce the variance, we focus on minimizing the bias and then perform the bandwidth selection for MSE reduction.

# 7 Closed Form Solution of Eigenvectors

The eigenvectors and the eigenvalues of the matrix $B = \nabla p(\mathbf{x})\nabla^\top y(\mathbf{x}) + \nabla y(\mathbf{x})\nabla^\top p(\mathbf{x})$ can be obtained in a closed form solution: the eigenvectors are

$$\mathbf{u}_1 = \frac{1}{\sqrt{2(1+\cos\theta)}}\left(\frac{\nabla p(\mathbf{x})}{||\nabla p(\mathbf{x})||} + \frac{\nabla y(\mathbf{x})}{||\nabla y(\mathbf{x})||}\right), \quad \text{and} \tag{79}$$

$$\mathbf{u}_2 = \frac{1}{\sqrt{2(1-\cos\theta)}}\left(\frac{\nabla p(\mathbf{x})}{||\nabla p(\mathbf{x})||} - \frac{\nabla y(\mathbf{x})}{||\nabla y(\mathbf{x})||}\right), \tag{80}$$

with corresponding eigenvalues

$$\lambda_1 = \frac{||\nabla p||||\nabla y(\mathbf{x})||}{2p(\mathbf{x})}(\cos\theta + 1) = \frac{1}{2p(\mathbf{x})}(\nabla y^\top\nabla p + ||\nabla y||||\nabla p||), \quad \text{and} \tag{81}$$

$$\lambda_2 = \frac{||\nabla p||||\nabla y(\mathbf{x})||}{2p(\mathbf{x})}(\cos\theta - 1) = \frac{1}{2p(\mathbf{x})}(\nabla y^\top\nabla p - ||\nabla y||||\nabla p||), \tag{82}$$

where $\theta$ is the angle between two gradients $\nabla p(\mathbf{x})$ and $\nabla y(\mathbf{x})$.

The sum of the two eigenvalues is the same as the bias. These explicit solutions can also be used for a faster calculation for metric $A_{NW}$. In our work, $\nabla y(\mathbf{x})$ is a constant vector due to a Gaussian model, and only $\nabla p(\mathbf{x})$ changes over points.

One particular point is where either $\nabla y(\mathbf{x})$ or $\nabla p(\mathbf{x})$ is zero. At this point, the estimation of the NW estimator is unbiased regardless of the choice of metric. This happens when $\mathbf{x}$ and $y$ are uncorrelated ($\Sigma_{y\mathbf{x}} = 0$) or when we estimate the density at the center $\mathbf{x} = \mu_{\mathbf{x}}$. In these two special cases, metric learning does not necessarily improve the leading terms of the bias.

## 8 Derivation of WMetric for Experiments

Our derivation of the gradient is slightly different from the derived result in [21], and we have applied our gradient in our experiment.

WMetric in [21] is a single global metric for all different points. The metric is obtained with minimization of the empirical MSE:

$$R = \sum_i (y_i - \widehat{y}_i)^2, \tag{83}$$

where $\widehat{y}_i = \widehat{y}(\mathbf{x}_i) = \frac{\sum_{j \neq i} y_j K(\mathbf{x}_j, \mathbf{x}_i)}{\sum_{j \neq i} K(\mathbf{x}_j, \mathbf{x}_i)}$ is the NW estimator with the metric matrix $A = LL^\top$. The derivative of $R$ with respect to $L$ gives

$$\frac{\partial R}{\partial L} = \sum_i 2(y_i - \widehat{y}_i)\left(-\frac{\partial \widehat{Y}_i}{\partial L}\right), \tag{84}$$

with additional derivatives

$$\frac{\partial \widehat{y}_i}{\partial L} = \frac{1}{K(\mathbf{x}_i, \mathbf{x}_j)}\sum_{j \neq i}(y_j - \widehat{y}_i)\frac{\partial K(\mathbf{x}_i, \mathbf{x}_j)}{\partial L}, \quad \text{and} \tag{85}$$

$$\frac{\partial K(\mathbf{x}_i, \mathbf{x}_j)}{\partial L} = -K(\mathbf{x}_i, \mathbf{x}_j)\frac{1}{\sigma^2}(\mathbf{x}_i - \mathbf{x}_j)(\mathbf{x}_i - \mathbf{x}_j)^\top L, \tag{86}$$

yielding a gradient of

$$\frac{\partial R}{\partial L} = \frac{2}{\sigma^2}\Big(\sum_i \frac{y_i - \widehat{y}_i}{\sum_{j \neq i} K(\mathbf{x}_i, \mathbf{x}_j)}\sum_{j \neq i}(y_j - \widehat{y}_i)K(\mathbf{x}_i, \mathbf{x}_j)\cdot(\mathbf{x}_i - \mathbf{x}_j)(\mathbf{x}_i - \mathbf{x}_j)^\top\Big)L. \tag{87}$$

Using the gradient, the transformation matrix $L$ can be obtained, and the global metric matrix can be obtained from $A = LL^\top$. The correctly derived gradient is slightly different from the equation presented in the original literature [21]. We used the gradient derived here in Eq. (87) for the "WMetric" in our experiments.

The calculation of the gradient is very costly because of the double summation of the matrices. In Eq. (87), both summations are with respect to the data indexes, and the calculation of one gradient scales with $N^2$.