[Reviews · NeurIPS 2017]

Reviewer 1



This paper follows the idea of “Generative local metric learning for nearest neighbor classification”, and analyze how metric can help for kernel regression with Nadaraya-Watson (NW) estimator. Although linear transformation (metric) has no impact on NW-kernel regression when the number of instances goes to infinity, it will reduce the bias given finite samples. The authors analyze that metric learning can significantly reduce the mean square error (MSE) in kernel regression, particularly for high-dimensional data. Together with a generative model, the method combines both the global and local information together. Experiments on both synthetic and real datasets show the effectiveness of the proposed method. The whole paper is well-written and easy to follow. Analyses, discussion, and comparisons of the proposed method and its relativeness with other methods are clear. Two suggestions on the paper: 1. Fig.1 gives a direct illustration of the impact of metric in kernel regression. There need more statements on the effect of \nabla y. 2. There needs a generative model to estimate the probability of data before computing the metric, and a single Gaussian model is used. Some of the following derivations are based on this assumption, the authors should make clear which computations are general and which are specific.

Reviewer 2



The idea of optimizing metric matrix used for NW regressor by explicitly counteracting the leading term of the bias of regressor is interesting. The (essentially) two-rank metric matrix construction is nice and experiments supports the proposed method well. My main concern is that two assumptions in Prop.2, 1) p(x) is away from zero, and 2) \nabla p(x) and \nabla y(x) are not parallel, are guaranteed or validated. The possibility of the violation of the assumption 2) could be negligible, but it is highly possible that p(x) is close to zero and the bias term diverges. Does the regularization in eq.(21) safely solves the problem ? Minor point: in supplementary material, eq.(11), the right most L^{-1} must be L^{T}. eqs.(11) and (12), subscript x for L should be removed.

Reviewer 3



Metric learning is one of the fundamental problems in person re-identification. This paper presents a metric learning method using Nadaraya-Watson (NW) kernel regression. The key feature of the work is that the NW estimator with a learned metric uses information from both the global and local structure of the training data. Theoretical and empirical 9 results confirm that the learned metric can considerably reduce the bias and MSE for kernel regression even when the data are not confined to Gaussian. The main contribution lies in the following aspects: 1. Provided a formation on how metric learning can be embedded in a kernel regression method. 2. Proposed and proved a theorem that guarantees the existence of a good metric that can eliminate the leading order bias at any point regardless of assumption of Gaussian distribution of the data. 3. Under Gaussian model assumption, provided an efficient algorithm that can guarantee to achieve an optimal metric solution for reducing regression error. Overall, this paper’s reasoning is rigorous. Description is clear. The experiments sufficiently demonstrate the author’s claims for the proposed method. However, it needs major revision before it is accepted. My comments are as follows: 1. The structure of the paper is confused. I suggest the authors add contributions of the papers in the introduction section. The usage and importance of this paper needs to be further clarified. 2. I also find some grammar problems in this paper. Author needs to carefully check these mistakes, which is very important for readers. 3. The limitations of the proposed method are scarcely discussed. 4. The works also cannot describe the latest developments since there are too few references after 2016. 5. The authors only show very few experiments results in section 5, I think it is not enough. Except MSE, the authors also should provide other comparisons, such as Root mean squared error of classication accuracy(RMSE) or MSE, etc. In this way, the results of the paper will be more reasonable.